# Trophoblast Cell Surface Antigen 2 (TROP2) as a Predictive Bio-Marker for the Therapeutic Efficacy of Sacituzumab Govitecan in Adenocarcinoma of the Esophagus

**DOI:** 10.3390/cancers14194789

**Published:** 2022-09-30

**Authors:** Sascha Hoppe, Lydia Meder, Florian Gebauer, Roland T. Ullrich, Thomas Zander, Axel M. Hillmer, Reinhard Buettner, Patrick Plum, Julian Puppe, Wolfram Malter, Alexander Quaas

**Affiliations:** 1Institute of Pathology, Faculty of Medicine and University Hospital Cologne, University of Cologne, 50937 Köln, Germany; 2Internal Medicine, Oncology Center for Integrated Oncology Aachen Bonn Cologne Duesseldorf, Department I of Internal Medicine, Faculty of Medicine and University Hospital Cologne, University of Cologne, 50937 Köln, Germany; 3Mildred Scheel School of Oncology Cologne, Faculty of Medicine and University Hospital Cologne, University of Cologne, 50937 Köln, Germany; 4Department of Gynecology and Obstetrics, Faculty of Medicine and University Hospital Cologne, University of Cologne, 50937 Köln, Germany; 5Department of General, Visceral, Cancer and Transplantation Surgery, Faculty of Medicine and University Hospital Cologne, University of Cologne, 50937 Köln, Germany

**Keywords:** TROP2, esophageal adenocarcinoma, sacituzumab govitecan, therapy response

## Abstract

**Simple Summary:**

The trophoblast cell surface antigen 2 (TROP2) is a protein produced by many carcinomas. Sacituzumab govitecan (SG) is a drug consisting of an antibody that binds to TROP2 on the tumor cell when TROP2 is present and is taken up into the cell interior after binding. The antibody is coupled with a cytotoxic substance (SN38) that is released inside the cell after uptake. As a result, a lethal dose of SN38 acts specifically in the tumor cell while having a small systemic effect, reducing the extent of side effects. We can show here that TROP2 is formed in nearly 90% of esophageal adenocarcinomas (EAC) and that sacituzumab govitecan is also effective in EAC. We can show that efficacy is dependent on the presence of TROP2 on the cancer cell - complete absence of TROP2 is associated with poor response rate to SG. We therefore advocate determining TROP2 at the protein level prior to therapy. Suitable immunohistochemical antibodies for routine testing exist.

**Abstract:**

Introduction: The Trophoblast cell surface antigen 2 (TROP2) is expressed in many carcinomas and may represent a target for treatment. Sacituzumab govitecan (SG) is a TROP2–directed antibody-drug conjugate (ADC). Nearly nothing is known about the biological effectiveness of SG in esophageal adenocarcinoma (EAC). Material and Methods: We determined the TROP2 expression in nearly 600 human EAC. In addition, we used the EAC cell lines (ESO-26, OACM5.1C, and FLO-1) and a xenograft mouse model to investigate this relationship. Results: Of 598 human EACs analyzed, 88% showed varying degrees of TROP2 positivity. High TROP2 positive ESO-26 and low TROP2 positive OACM5.1C showed high sensitivity to SG in contrast to negative FLO-1. In vivo, the ESO-26 tumor shows a significantly better response to SG than the TROP2-negative FLO-1 tumor. ESO-26 vital tumor cells show similar TROP2 expression on all carcinoma cells as before therapy initiation, FLO-1 is persistently negative. Discussion: Our data suggest that sacituzumab govitecan is a new therapy option in esophageal adenocarcinoma and the TROP2 expression in irinotecan-naïve EAC correlates with the extent of treatment response by sacituzumab govitecan. TROP2 is emerging as a predictive biomarker in completely TROP2-negative tumors. This should be considered in future clinical trials.

## 1. Introduction

Trophoblast cell surface antigen 2 (TROP2) is a 323 amino acid transmembrane glycoprotein that is a calcium signal transducer and is responsible for tumor cell growth in malignant tumors [1,2,3]. TROP2 is expressed in various malignant tumors, including breast carcinomas and pancreatic carcinomas [4]. TROP2 is also physiologically expressed in the placenta and during fetal lung development, among others [5,6,7]. Sacitumzumab govitican (SG) is the first antibody-drug conjugate in which the biologically active metabolite of the topoisomerase 1 inhibitor irinotecan, SN-38 is coupled via a linker to the humanized antibody hRS7 IgG1, which binds to trophoblast antigen 2 (TROP2) [8].

This makes TROP2 an ideal therapeutic target protein. In a phase-3 clinical trial on metastatic, triple-negative breast carcinoma (ASCENT), the outstanding clinical effectiveness of this form of therapy was demonstrated. The overall survival was doubled compared to the chemotherapy arm (investigator choice). These results led to full approval of SG (Trodelvy^®^) by the FDA for triple negative metastatic breast cancer in April 2021 [9,10].

The clinical efficacy of sacitumzumab govitican (SG) has also been demonstrated in other solid tumors expressing TROP2 [8].

However, it was already apparent in the ASCENT study that not all patients benefited equally from SG therapy—see also below. Little is known about the predictors of response to SG therapy. This question was addressed in a recently published study in breast cancer, which investigated mechanisms of de novo and acquired resistance to SG therapy [11].

One finding of this study was that the tumor of a patient who showed no response to the application of Trodelvy^®^ (de-novo progression) showed no mRNA or protein expression of TROP2 (tumor cells completely negative for TROP2). In contrast, other patients with good therapeutic responses showed detectable TROP2 expression in their tumor cells.

These results confirm the findings of a previous study that the lack of TROP2 expression on tumor cells leads to de novo clinical resistance to SG [9].

The more detailed analyses of the ASCENT study also fit in with this. Patients with high or medium TROP2 expression had significantly higher response rates to SG than tumors with low TROP2 expression (44% (high), 39% (medium), 22% (low)). Six out of seven patients who showed no TROP2 expression at all on their tumor cells also showed no therapeutic response to SG [10].

Very few data exist on the effectiveness of SG in metastatic esophageal cancer [8]. This is relevant insofar as irinotecan is not one of the agents usually used in the early lines of therapy or in neoadjuvant treatment of esophageal carcinoma. The charm of SG application compared to the significantly cheaper irinotecan lies in the better side effect profile of SG treatment.

Our study was the first to investigate the actual frequency of TROP2 expression on carcinoma cells in a very large number of esophageal adenocarcinomas (EAC) (n = 598). Furthermore, the therapeutic efficacy of sacitumzumab govitican (SG) was investigated in different EAC cell lines with different TROP2 expression levels and in a corresponding xenograft mouse model.

## 2. Methods and Material

### 2.1. Cell Lines

Human esophageal adenocarcinoma (EAC) cell lines Eso26 and OACM5.1C were obtained from Deutsche Sammlung von Mikroorganismen und Zellkulturen (DSMZ, Braunschweig, Germany). Flo-1 was kindly provided by Winand N.M. Dinjens, Department of Pathology, Erasmus MC, University Medical Center Rotterdam. All cell lines were cultivated in RPMI-1640 (Life Technologies, Carlsbad, CA, USA) supplemented with 10% fetal bovine serum (Capricorn Scientific, Ebsdorfergrund, Germany), 1% penicillin (Life Technologies, USA), and 1% streptomycin (Life Technologies, USA) in a humidified atmosphere of 5% CO_2_ at 37 °C. All cultures were tested for contamination of mycoplasms by qualitative PCR.

### 2.2. RNA Analysis

RNA extraction was performed using the AllPrep DNA/RNA Mini Kit (Qiagen, Venlo, Germany). Libraries for RNA next-generation sequencing were generated by TruSeq mRNA (Illumina) and sequenced on an Illumina NovaSeq 6000 with 2 × 150 bp and 30 Mio read pairs per sample. Reads were aligned to the human genome (*Homo sapiens* GRCh38) using STAR software v. 2.6 [12,13].

Mapped reads were counted with HTSeq [13]. Read counts were normalized with DESeq2 for sequencing depth and RNA composition using the median of ratios method. [14].

A complete analysis of transcriptomic data will be published elsewhere (manuscript in preparation).

### 2.3. Chemosensitivity Assay

Cell viability under treatment was determined on a 96-well-plate with 2500 cells/well seeded in triplicates 24 h prior to treatment. For a series of dilutions, SN-38 (MedChemExpress LLC, Monmouth, NY, USA) or Trodelvy (Sacituzumab govitecan-hziy, Immunomedics Inc., Morris Plains, NY, USA) was first dissolved in 0.9% sodium chloride (NaCl) and equal volumes were further diluted with standard growth media. After 72 h of treatment cell viability was determined via CellTiter-Glo^®^ 2.0 Cell Viability Assay (Promega GmbH, Walldorf, Germany) according to the manufacturer’s protocol. Luminescence was measured with a microplate reader Centro LB 960 (Berthold Technologies GmbH & Co. KG, Bad Wildbad, Stuttgart, Germany). All experiments were repeated at least three times.

### 2.4. Immunocytochemistry

Human EAC cell lines were detached from the cell culture vessel by trypsinization, centrifuged (5 min, 1000 rpm) and resuspended in PBS, centrifuged (5 min, 1000 rpm), and fixated in 4% PFA at 4 °C over-night. Fixated cells were centrifuged (5 min, 1000 rpm) and resuspended in 97% ethanol and 3 drops of protein glycerine (Carl Roth GmbH, Karlsruhe, Germany). After another centrifugation (5 min, 1000 rpm) the supernatant was decanted and the cell pellet was transferred to a biopsy cassette for formalin embedding with the automated instrument Technicon.

Xenograft tissue was fixated in 4% PFA overnight and further processed with the automated tissue processor Technicon. Xenograft sections were stained with hematoxylin and eosin (H&E).

### 2.5. In Vivo Xenograft Treatment

Six-to-eight-week-old NSG mice (NOD.Cg-PrkdcscidIl2rgtm1Wjl/SzJ) purchased from CECAD (Cluster of Excellence—Cellular Stress Responses in Aging-Associated Diseases, Cologne, Germany) were anesthetized with isoflurane for controlled injection of cells. Human xenografts were generated by subcutaneous injection of 5*106 human EAC cells in 100 µL PBS bilateral into both flanks of NSG mice. Tumor volume was measured by caliper and calculated with the formula V = (Length × Width^2^)/2. Mice were randomly distributed into groups and treatment was started when a xenograft reached a volume of 50 mm^3^. Mice were treated on day 0 (therapy initiation), day 3, day 7, and day 10 with 25 mg/kg/body weight (BW) Trodelvy (Sacituzumab govitecan-hziy, Immunomedics Inc., USA) or with solvent-control by i.p. injection in concordance with the established protocol for breast cancer xenograft treatment [15]. The total treatment time was 10 days. All animal experiments were approved by the local Ethics Committee of Animal experiments.

### 2.6. Patient Cohort

We analyzed formalin-fixed, paraffin-embedded material from 598 patients with EAC who underwent primary surgical resection or resection after neoadjuvant therapy between 1999 and 2016 at the Department of General, Visceral and Cancer Surgery, University of Cologne, Germany. The standard surgical procedure was laparotomic or laparoscopic gastrolysis and right transthoracic en bloc esophagectomy including two-field lymphadenectomy of mediastinal and abdominal lymph nodes. Reconstruction was performed by high intrathoracic esophagogastrostomy as described previously [16].

Patients with advanced esophageal cancer (cT3, cNx, M0) received preoperative chemoradiation (5-FU, cisplatin, 40 Gy as treated in the area prior to the CROSS trial) or chemotherapy alone. All patients were followed up according to a standardized protocol. During the first 2 years, patients were followed up clinically in the hospital every 3 months. Afterward, annual exams were carried out. Follow-up examinations included a detailed history, clinical evaluation, abdominal ultrasound, chest X-ray, and additional diagnostic procedures as required. Follow-up data were available for all patients. Patient characteristics are given in Table 1. Depending on the effect of neoadjuvant chemo- or radio-chemotherapy, there is a preponderance of minor responders in the TMAs, defined as histopathological residual tumor of ≥10% [17].

We used the format of tissue microarrays (TMAs) consisting of 598 patients with EAC (see Table 1).

For tissue microarray analysis (TMA), one tissue core from each tumor was punched out and transferred into a TMA recipient block. TMA construction was performed as previously described [18,19].

In brief, tissue cylinders with a diameter of 1.2 mm each were punched from selected tumor tissue blocks using a self-constructed semi-automated precision instrument and embedded in empty recipient paraffin blocks. Then, 4-μm sections of the resulting TMA blocks were transferred to an adhesive-coated slide system (Instrumedics Inc., Hackensack, NJ, USA) for immunohistochemistry.

To get a better feeling of how the antibodies used stain already known TROP2 expressing carcinomas (such as breast carcinomas) and also normal tissues we have studied 50 breast carcinomas, 50 pancreatic carcinomas, and 5 each of normal squamous esophageal mucosa and Barrett’s mucosae (Appendix A).

### 2.7. Immunohistochemistry

Immunohistochemistry (IHC) was performed on TMA slides. TROP2 showed a membrane staining pattern. TROP2 immunohistochemistry was performed by a laboratory-developed test (LDT) using primary antibody clone ERP20043 (abcam, rabbit monoclonal) at 1:1000 dilution on the Leica Bond Max staining platform (Leica Biotechnologies, Wetzlar, Germany). Antigen-retrieval was achieved by heat-induced antigen-retrieval with EDTA buffer. Detection was done by the Bond Polymer Refine system (Leica Biotechnologies). Additionally, we used the Trop-2 mouse monoclonal antibody (ENZ ABS380-0100; ENZO Life Sciences, Farmingdale, NY, USA) on the Ventana benchmark as used in the ASCENT study (1:100 dilution with heat-induced antigen-retrieval with EDTA buffer). We have analyzed all tumors with both antibodies. Tonsil tissue was used as on-slide control on all stainings. We evaluated all tumors according to the H-score and converted the results into a four-level scheme (0–3, compare Table 1). Here, 0 = the absence of TROP2 expression on tumor cells, 1 = an H-score expression level of 1–100, 2 = 101–200, and 3 = 201–300.

### 2.8. Data Analysis and Statistics

Statistical analysis of the chemosensitivity assays was conducted using GraphPad Prism 5. Data are presented as means ± SEM. Relative luminescence units (RLU) were normalized to untreated cells. Non-linear regression curves were calculated by the equation Sigmoidal, 4PL, x is concentration (Y = Bottom + (Top − Bottom)/(1 + (IC50/X)^HillSlope)) from GraphPad Prism to determine the half maximal inhibitory concentration (IC_50_ values). The top values of the regression curve were constrained to be constantly equal to 100. For sets of data that were not appropriate for the equation, the IC_50_ value was estimated manually (Flo-1 treated with SG).

Descriptive analysis included the frequency of nominal parameters, the median with lower (LQ) and upper (UQ) quartiles for numeric variables (ordinal or asymmetric distribution), and the mean for numeric variables with a normal distribution. Univariate analysis was conducted for tables using chi-squared statistics or Fisher’s exact test if necessary. The Cochran–Armitage test for trend was used to determine whether there was a linear trend between the column number and the fraction of subjects in the top row. The Mann–Whitney U test was used to compare continuous variables. Significant differences between groups were defined as *p* < 0.05.

The prognosis was calculated including all types of mortality beginning on the date of surgery. Kaplan–Meier univariate analysis was used to describe survival distribution, and log-rank tests were used to evaluate survival differences. Cox proportional hazard regression with sequential backward elimination of the non-significant variables was used to analyze the effect of several risk factors on survival. Survival analysis and the multivariate cox-regression model were used on the entire cohort (test + validation cohort).

Statistical analyses were carried out using IBM SPSS v22.0 (IBM Corporation, New York, NY, USA). MedCalc Statistical Software version 18.2.1 (MedCalc Software bvba, Ostend, Belgium; http://www.medcalc.org; accessed on 1 February 2018) was used for the graphic presentation of the results.

## 3. Results

### 3.1. Patients’ Baseline Characteristics

On the TMA, a total of 598 patients were immunohistochemically interpretable for TROP2 (Table 1). Patients were predominantly men (male n = 531, 88.2%, female n = 67, 11.2%). The median age of the entire patient cohort at the time of diagnosis was 65.2 years (range 33.6–85.6 years). In 351 patients (58.7%), a neoadjuvant treatment (chemo- or radiochemotherapy) was performed before surgery.

### 3.2. TROP2 Is Expressed by the Majority of EAC Tumors

Remarkably, expression of TROP2 was detectable in 526 patients (88.0%) (Table 1).

Tumors expressing TROP2 were found to be differentially strong and homogeneous. Approximately 66% of the tumors exhibited largely homogeneous (>80% stained tumor cells) and moderate to strong membranous labeling of TROP2. 22% of tumors had only weak membranous expression, barely or not at all detectable in the overview (score 1) and 12% of tumors were TROP2 negative (score 0) (Appendix A). A correlation between TROP2 and any clinical or histopathological parameter could not be found in cross-table (chi-square test) analysis. TROP2 expression in EAC is not associated with tumor stage, presence of lymph node metastasis, or UICC stage.

Both antibodies used show almost identical staining results. The mouse monoclonal antibody has a slight tendency to overstain with partial membranous and cytoplasmic labeling. Together, TROP2 expression analysis of our large cohort of EAC patients demonstrates a high prevalence of Trop2 positivity in esophageal adenocarcinoma.

### 3.3. Sacituzumab Govitecan Specifically Inhibits TROP2 Positive EAC Cells

Established esophageal adenocarcinoma cell lines Eso26, OACM 5.1C and Flo1 were used as a model to demonstrate the relevance of TROP2 expression levels to the effectiveness of treatments with sacituzumab govitecan (SG). Analysis of the TROP2 expression in these cells showed that Eso26 homogenously expresses high levels of TROP2 mRNA and protein, while Flo1 lacks TROP2 expression. OACM 5.1C shows a low-level mRNA expression as well as just a few TROP2-positive tumor cells (Figure 1A,B). The dilution series of SG showed a ~15-fold decreased response for the TROP2 negative cell line Flo1 compared to TROP2 positive Eso26 with IC50-values of ~100 nM and 6.5 nM, respectively (Figure 1C). SN-38, the active compound of the antibody–drug conjugate SG alone, has little effect equally on both cell lines (Flo1 = IC50 ~100 nM; Eso26 = IC50 97.4 nM), demonstrating the protein-dependence of SG. The FLO-1 cell line has a special delayed growth behavior with rapid progression compared to the early onset Eso26 with a similar rapid progression. The delayed tumor onset followed by rapid tumor growth of FLO-1 is in concordance with the literature. In the study by Lui DS and colleagues, tumor growth of the parental FLO-1 starts lately, around day 80 followed by a rapid increase in tumor mass [20]. For the results of the in vitro and in vivo response to irinotecan please compare Appendix A. Together, our in vitro experiments show differential efficacy of SG to inhibit EAC cell lines with a positive correlation to TROP2 expression levels.

### 3.4. TROP2 Positive EAC Tumors Are Sensitive to Sacituzumab Govitecan

We generated subcutaneous xenografts in NSG mice with the cell lines Eso26 (TROP2+) and Flo1 (TROP2-). After sufficient tumor growth, mice were treated with either SG or vehicle solution for 10 days. Strikingly, SG showed no effect on TROP2 negative tumors (FLO-1, size after treatment 524.6 mm^3^, negative control 697.3 mm^3^) while significantly decreasing the growth of TROP2 positive tumors (Eso26, size after treatment 27.9 mm^3^, negative control 597.4 mm^3^, Figure 2A,B). Of note, TROP2 negative Flo1 tumors grew much slower than positive Eso26 tumors. Dissected tumors after treatment were analyzed for TROP2 expression by IHC and showed persistence of uniform positive or negative TROP2 expression throughout the treatment. In the frame of health monitoring, we routinely determine the body weight of mice. We have not detected any toxicity (Appendix A).

## 4. Discussion

Our experimental data show that sacitumzumab govitican (SG) also exerts highly effective tumor suppressive properties in esophageal adenocarcinoma. The biological effectiveness of SG requires the presence of TROP2 on the tumor cell, although the level of expression is apparently less significant. Only the complete absence of TROP2 on tumor cells is associated with a significantly worse response rate of SG. For example, the TROP2 negative cell line FLO-1 shows only a minimal response rate to SG. We were able to demonstrate the sensitivity of FLO-1 to irinotecan in the control arm of our analyses. The fact that a low response rate was detectable at all is presumably not mediated by TROP2-dependent internalization of the cytotoxic agent SN-38, but by delivery of SN-38 into the intercellular space, such that SN-38 induces immediate, TROP2-independent cytotoxicity. This mechanism of SG has been described and is explained on the basis of a model detachment of SN-38 from its linker [21]. However, the extent of direct cell damage is small, so the presence of TROP2 on the tumor cells as a prerequisite for effective internalization of SN-38 induces the actual relevant tumor-suppressing effect. Our data is consistent with results in breast carcinoma [10,11]. These studies also showed an almost linear correlation of the TROP2 expression level with the therapeutic efficacy of SG. Patients with high or medium TROP2 expression in the ASCENT study had significantly higher response rates to SG than tumors with low TROP2 expression (44% (high), 39% (medium), 22% (low)). We were able to show in a very large tumor cohort (n = 598) that adenocarcinomas of the esophagus are very frequently TROP2 positive (88%). The majority of cases (65.7%) showed medium to high and homogeneous TROP2 expression. Only in 12% of our cohort was TROP2 undetectable. In these tumors, we would expect minimal to no clinical response to SG. In breast carcinoma, six out of seven TROP2 negative carcinomas showed no response to SG [10]. This makes the EAC overall a potentially SG highly sensitive tumor entity. We and others found a very good correlation between mRNA and protein expression of TROP2 [11] so a simple immunohistochemical analysis with a suitable antibody allows reliable statements on the actual TROP2 status of the tumor. This makes TROP2 an easily determinable therapy-relevant biomarker. In the only clinical basket study currently available, which included 495 patients with different tumor entities, the effectiveness of SG was also investigated in 19 esophageal carcinomas of different histology (including squamous cell carcinomas) [8]. In this IMMU-132-01 trial, a low response to SG was observed in esophageal cancer. This is probably due to previously used irinotecan and related to resistance in the surviving clones or due to the absence of TROP2, which was not evaluated in the study. The vast majority of operable esophageal carcinomas are treated neoadjuvantly at present. As a rule, irinotecan-free chemotherapy combinations are used. In the case of tumor recurrence, the tumor clones are irinotecan-naïve. The use of SG will thus be considered in a large proportion of patients with EAC and could be an effective treatment option. In particular, patients who have tolerated neoadjuvant poorly could benefit from the comparatively low toxic SG treatment compared to systemic irinotecan administration.

## 5. Conclusions

In conclusion, the results of our data suggest that sacituzumab govitecan is a new therapy option in esophageal adenocarcinoma. The TROP2 expression correlates with the extent of treatment response by sacituzumab govitecan. We argue for the establishment of TROP2 determination on carcinoma cells as a predictive biomarker. At least in tumors that are completely TROP2 negative, the therapeutic efficacy of SG appears to be significantly worse in both breast (according to re-analyses of the ASCENT-study) and esophageal cancer (according to our own data). This needs to be addressed in future clinical trials.

## Figures and Tables

**Figure 1 cancers-14-04789-f001:**
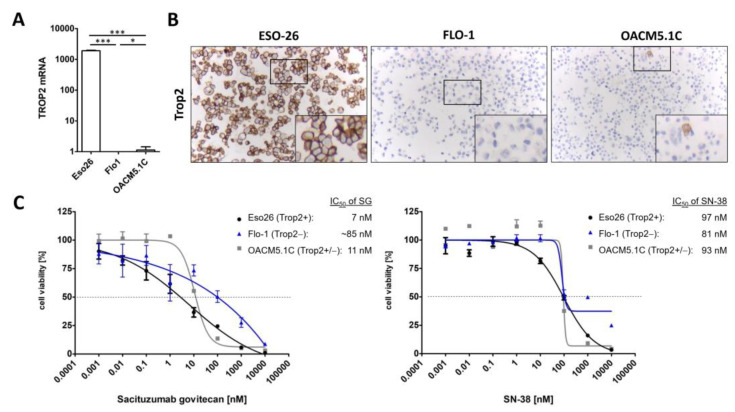
Transcript counts of TACSTD2, the gene for TROP2, from RNA-seq data normalized for sequencing depth and RNA composition using the median of ratios method (**A**) and protein expression by IHC staining with anti-TROP2 antibody (**B**, ab214488, abcam, 200×) of EAC cell lines ESO-26, OACM5.1C and FLO-1 in vitro. (**C**) Viability assay of Eso-26, OACM5.1C, and Flo-1 treated with concentrations of Sacituzumab govitecan (**left**) or SN-38 (**right**), detected by CellTiterGlo 2.0 (Promega). Values are presented relative to vehicle-treated control as means ± SEM with a non-linear regression curve to determine the IC50, n = 3, * *p*-value < 0.05, *** *p*-value < 0.001, two-sided unpaired Student’s *t*-test.

**Figure 2 cancers-14-04789-f002:**
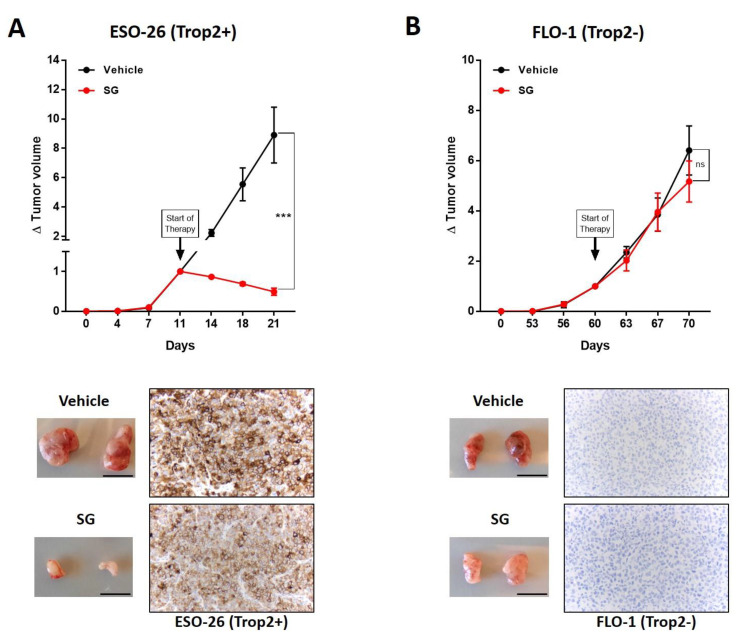
Subcutaneous xenografts of ESO-26 (**A**) and FLO-1 (**B**) were treated with SG or vehicle. Tumor volume relative to the volume at therapy initiation (∆ tumor volume) is presented as means ± SEM, n = 4. Macroscopic view of the dissected tumors (bar = 1 cm) next to IHC stains with anti-TROP2 antibody (200×) after vehicle or SG treatment is shown below, *** *p*-value < 0.001, two-sided unpaired Student’s *t*-test.

**Table 1 cancers-14-04789-t001:** The protein expression of TROP2 was determined by immunohistochemistry in 598 patients with operable adenocarcinoma of the esophagus (EAC). 88% of the tumors showed varying degrees of TROP2 positivity (1 weak expression, 3 strong and homogeneous expression). Almost 60% of these tumors were treated neoadjuvantly (according to CROSS or chemotherapy alone).

				TROP2 Expression								
				0		1		2		3		*p* Value
**total**		598		72	12.0%	133	22.2%	268	44.8%	125	20.9%	
**sex**	female	67	11.2%	13	19.4%	19	28.4%	23	34.3%	12	17.9%	
	male	531	88.8%	59	11.1%	114	21.5%	245	46.1%	113	21.3%	0.077
**agegroup**	<65 yrs	310	51.9%	33	10.6%	74	23.9%	142	45.7%	62	19.8%	
	>65 yrs	288	48.1%	38	13.2%	58	20.2%	128	44.5%	63	22.1%	0.554
**Neoadjuvant**	No	247	41.3%	34	13.8%	53	21.5%	104	42.1%	56	22.7%	
**treatment**	Yes	351	58.7%	38	10.8%	80	22.8%	164	46.7%	69	19.7%	0.480
**Tumor stage**	pT1	114	19.1%	12	10.5%	29	25.4%	55	48.2%	18	15.8%	
	pT2	114	19.1%	14	12.3%	21	18.4%	52	45.6%	27	23.7%	
	pT3	349	58.6%	43	12.3%	78	22.3%	154	44.1%	74	21.2%	0.828
	pT4	19	3.2%	2	10.5%	4	21.1%	7	36.8%	6	31.6%	
**Lymph node**	pN0	234	39.3%	33	14.1%	60	25.6%	102	43.6%	39	16.7%	
**metastasis**	pN1	197	33.1%	22	11.2%	44	22.3%	90	45.7%	41	20.8%	
	pN2	86	14.4%	9	10.5%	10	11.6%	42	48.8%	25	29.1%	0.173
	pN3	79	13.3%	8	10.1%	19	24.1%	33	41.8%	19	24.1%	
**UICC stage**	I	85	14.3%	9	10.6%	22	25.9%	39	45.9%	15	17.6%	
	II	73	12.2%	10	13.7%	20	27.4%	32	43.8%	11	15.1%	
	III	275	46.1%	35	12.7%	61	22.2%	124	45.1%	55	20.0%	0.508
	IV	163	27.3%	17	10.4%	29	17.8%	73	44.8%	44	27.0%	

## Data Availability

The datasets generated during and/or analysed during the current study are available from the corresponding author on reasonable request.

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
