# Peer review of "Trophoblast Cell Surface Antigen 2 (TROP2) as a Predictive Bio-Marker for the Therapeutic Efficacy of Sacituzumab Govitecan in Adenocarcinoma of the Esophagus"

_cancers, 2022, doi:10.3390/cancers14194789_

Round 1

Reviewer 1 Report

The authors present important and novel evidence regarding the potential application of the ADC, sacituzumab govitecan, to EACs, a disease in need of additional effective therapeutic options. The authors demonstrate a range of TROP2 expression in a large number of human EAC tumor samples, of which the vast majority show at least some TROP2 expression, and they show that TROP2 expression is not associated with cliniopathologic features.  The authors have generated consistent in-vitro and in-vivo evidence of “on-target” SG activity in EAC cell line models, including the highly encouraging demonstration of the shrinkage of TROP2-high ESO26 xenograft tumors, suggesting that TROP2 (the target protein for the SG antibody) can function as a predictive biomarker for SG therapies in EAC. 

Enthusiasm for the novel and important findings is somewhat diminished due to uncertainty about the methods and lack of context for the results of the IHC / TMA. Another issue is that some of the conclusions lack clear support from the results.

Major issues:

1)    One key question regarding ADC in general is whether the tumor expresses the target at a high level, in particular when compared with normal tissues.  It is unclear from the data presented or the references cited if EAC expression of TROP2 is higher than normal tissues, or alternatively, if TROP2 expression in EAC is comparable to other high expressing cancers.  For example, squamous esophagus and skin is typically high in TROP2 expression. How does TROP2 expression in EAC compare to the normal tissues?  Alternatively, how does TROP2 expression in EAC compare to cancers where TROP2 is known to be highly expressed (and SG is known to be effective, ie TNBC)?

Since RNA and protein expression correlate well, a case could be made based on RNA expression data where available (in the absence of protein level data).

2)    A related concern has to do with the description of the methods for the TMA.  The methods state H-score was used, but staining is reported as 0-4 scale.  Can the scoring system be clarified?  Similarly, the methods describe two different antibodies that were used for IHC in the study.  Were all samples tested with 2 different antibodies (the use of 2 different antibodies for the same samples would be a real strength that should be more clearly highlighted)?  If so, show are the results from the two different antibodies reconciled? Otherwise, if some samples were tested with one antibody and other samples with another, what proportion of samples were tested with which antibody?

3)  The conclusion section is not consistently supported by the results or literature, as currently indicated in the article, and some portions are a bit unclear.  A revised version should address the following points:

- Line 266: “Only the complete absence of TROP2 on tumor cells is associated with a significantly worse response rate of SG”; is this based on Fig 1C? If so the IC50 should be stated for OACM5.1.

- Line 268: “We were able to demonstrate the sensitivity of FLO-1 for irinotecan in a control arm of our analyses” I do not find the data for irinotecan in FLO1.   Further, Line 234, suggests the active metabolite of irinotecan actualy has “little effect equally on both cell lines [eso26 and Flo1]”, rather than being effective.  Please clarify.

-  While Line 294 indicates one likely possibility for lack of responses to SG in esophageal cancers without mentioning other possibilities, such as the TROP2 expression is not high in EAC or ESCC compared to normal tissues or to other sensitive cancers (this is currently not clearly indicated).

Minor issues:

- Line 42; Reference 2 refers to published safety and efficacy results from a clinical trial and does not show data related to TROP2 expression in tumor vs normal cells, as currently stated.  Please clarify the reference.

-Non-standard FLO1 culture are used (RPMI instead of DMEM), would mention and / or explain.

-section 3.3 (or figure 1) could be improved by stating the IC50 values for OACM5.1 along with the data for other 2 cell lines

-Figure 1 C, Cell titer glo measures cell viability, rather than survival; would recommend clarifying this in the figure and text.

-Reference 2 and 6 appear to be identical.

-Line 300, would temper enthusiasm to not say SG is “very likely” effective; although the results of this study are indeed encouraging.

-Starting in line 303, would suggest to clarify which statements are based on clinical data, and which are based on pre-clinical data.

Author Response

Point-by-point answers to Reviewer-Comments: Revised Manuscript: “Trophoblast cell surface antigen 2 (TROP2) as a predictive biomarker for the therapeutic efficacy of sacituzumab govitecan in adenocarcinoma of the esophagus”

We would like to thank our reviewers for their helpful comments and suggestions for improvements. We have made every effort to answer your questions point by point and have repeatedly included aspects in the manuscript. Our answers and additions to the manuscript are highlighted in yellow. We have clarified the primary figures and added various other figures (supplementary figures 1-5). We are convinced that our manuscript could be significantly improved.

Reviewer 1

The authors present important and novel evidence regarding the potential application of the ADC, sacituzumab govitecan, to EACs, a disease in need of additional effective therapeutic options. The authors demonstrate a range of TROP2 expression in a large number of human EAC tumor samples, of which the vast majority show at least some TROP2 expression, and they show that TROP2 expression is not associated with cliniopathologic features.  The authors have generated consistent in-vitro and in-vivo evidence of “on-target” SG activity in EAC cell line models, including the highly encouraging demonstration of the shrinkage of TROP2-high ESO26 xenograft tumors, suggesting that TROP2 (the target protein for the SG antibody) can function as a predictive biomarker for SG therapies in EAC. 

Enthusiasm for the novel and important findings is somewhat diminished due to uncertainty about the methods and lack of context for the results of the IHC / TMA. Another issue is that some of the conclusions lack clear support from the results.

Major issues:

1)    One key question regarding ADC in general is whether the tumor expresses the target at a high level, in particular when compared with normal tissues.  It is unclear from the data presented or the references cited if EAC expression of TROP2 is higher than normal tissues, or alternatively, if TROP2 expression in EAC is comparable to other high expressing cancers.  For example, squamous esophagus and skin is typically high in TROP2 expression. How does TROP2 expression in EAC compare to the normal tissues?  Alternatively, how does TROP2 expression in EAC compare to cancers where TROP2 is known to be highly expressed (and SG is known to be effective, ie TNBC)?

Since RNA and protein expression correlate well, a case could be made based on RNA expression data where available (in the absence of protein level data).

Thank you very much for this significant comment. We now consider the results of a meta-analysis on the topic. This work shows the expression level of TROP2 in different tumors and we further consider a work that looks at TROP2 during fetal development. To gain a better understanding of the expression level of TROP2 as measured by our immunohistochemical antibodies we have performed complementary analyses. We compared the protein expression level of TROP2 in EAC with the expression level in other malignant tumors (breast carcinoma (n=50, pancreatic carcinoma n=50) and with normal esophageal tissue. All of the mammary carcinomas studied (n=50) showed strong membranous expression in over 80% of tumor cells. Of the 50 pancreatic carcinomas studied, 48 also showed identical membranous TROP2 expression and thus comparable expression levels as in EAC or breast carcinoma (Supplemental Figure 4). We also analyzed expression differences in normal esophageal tissue and Barrett's mucosae. This shows that the normal squamous mucosa of the esophagus is strong and consistentlyTROP2 positive (only the immediate basal cell layer is more focally and weakly marked). The Barrett's mucosa is also TROP2 positive but the expression level is lower and more focal than in physiological squamous epithelium. No TROP2 expression is found in the surrounding stroma, muscularis, or endothelia (Supplemental Figure 3).

2)    A related concern has to do with the description of the methods for the TMA.  The methods state H-score was used, but staining is reported as 0-4 scale.  Can the scoring system be clarified?  Similarly, the methods describe two different antibodies that were used for IHC in the study.  Were all samples tested with 2 different antibodies (the use of 2 different antibodies for the same samples would be a real strength that should be more clearly highlighted)?  If so, show are the results from the two different antibodies reconciled? Otherwise, if some samples were tested with one antibody and other samples with another, what proportion of samples were tested with which antibody?

Thank you for the opportunity to clarify these important points. We evaluated all tumors according to the H-score and converted the results into a four-level scheme (0-3, compare Table 1). Here, 0=the absence of TROP2 expression on tumor cells, 1 = an H-score expression level of 1-100, 2=101-200, and 3=201-300. We added this aspect in the methods section. We used two different monoclonal antibody clones: 1) ENZ ABS380-0100 - this clone was used for retrospective TROP2 -determination on tumor cells in the ASCENT study and 2) ERP20043, a rabbit monoclonal antibody. Rabbit monoclonal antibodies often show very clear and excellent reproducible results. We already know the comparative analysis with different antibody clones from other biomarkers (like Her2/neu, PD-L1 or currently with Claudin 18.2). We have analyzed all tumors with both antibodies. You are absolutely right, we have now made this aspect clear in the manuscript. Both antibodies show identical analysis results - we have now also illustrated the similarity of the staining results in the supplement.

3)  The conclusion section is not consistently supported by the results or literature, as currently indicated in the article, and some portions are a bit unclear.  A revised version should address the following points:

- Line 266: “Only the complete absence of TROP2 on tumor cells is associated with a significantly worse response rate of SG”; is this based on Fig 1C? If so the IC50 should be stated for OACM5.1.

The IC50 Values have been added to Fig. 1C.

- Line 268: “We were able to demonstrate the sensitivity of FLO-1 for irinotecan in a control arm of our analyses” I do not find the data for irinotecan in FLO1.   Further, Line 234, suggests the active metabolite of irinotecan actually has “little effect equally on both cell lines [eso26 and Flo1]”, rather than being effective.  Please clarify.

Thank you for pointing this out. We added a supplementary figure 1 showing the response to irinotecan in Eso26 and FLO-1 tumors in vivo. Regarding line 234, we added “Fig. 1C” for additional clarity.

-  While Line 294 indicates one likely possibility for lack of responses to SG in esophageal cancers without mentioning other possibilities, such as the TROP2 expression is not high in EAC or ESCC compared to normal tissues or to other sensitive cancers (this is currently not clearly indicated).

We have gone into detail on this point as well - please compare our comments above (point1). We have performed supplementary analysis on further tumors and normal tissue there.

Minor issues:

- Line 42; Reference 2 refers to published safety and efficacy results from a clinical trial and does not show data related to TROP2 expression in tumor vs normal cells, as currently stated.  Please clarify the reference.

Indeed, another publication needs to be cited at this point.

-Non-standard FLO1 culture are used (RPMI instead of DMEM), would mention and / or explain.

RPMI is often used as an alternative to DMEM to culture Flo1. It is recommended by the DSMZ (https://www.dsmz.de/collection/catalogue/details/culture/ACC-698).

-section 3.3 (or figure 1) could be improved by stating the IC50 values for OACM5.1 along with the data for other 2 cell lines

We agree. The IC50 values have been added to Fig. 1.

-Figure 1 C, Cell titer glo measures cell viability, rather than survival; would recommend clarifying this in the figure and text.

We have changed the axis labeling accordingly in Fig. 1C.

-Reference 2 and 6 appear to be identical.

Reference has now been adjusted.

-Line 300, would temper enthusiasm to not say SG is “very likely” effective; although the results of this study are indeed encouraging.

We have adapted the wording: .”...... and could be an effective treatment option”.

-Starting in line 303, would suggest to clarify which statements are based on clinical data, and which are based on pre-clinical data.

Thank you very much - we have now made this clear: “In conclusion the results of our data suggest that sacituzumab govitecan is a new therapy option in esophageal adenocarcinoma. The TROP2 expression correlates with the extent of treatment response by sacituzumab govitecan. We argue for the establishment of TROP2 determination on carcinoma cells as a predictive biomarker. At least in tumors that are completely TROP2 negative, the therapeutic efficacy of SG appears to be significantly worse in both breast (according to re-analyses of the ASCENT-study) and esophageal cancer (according to our own data). This needs to be addressed in future clinical trials”.

Reviewer 2 Report

  •  
  •  
  •  
  •  
  •  
  •  
  •  
  •  
  • Although the manuscript is clear and generally well written, it presents some important points that must be improved before publication. In particular:
  •  
  • Introduction: The introduction on TROP2 must be implemented. In particular, authors must introduce the function of this protein and specify that it is not only expressed in esophagus adenocarcinoma but also in other types of cancers. Moreover, TROP2 is also expressed in non cancerous tissues (as reported in  PMID: 32726711, 21743029)

3.2. TROP2 is expressed by the majority of EAC tumors: representative images of TROP2 expression in EAC must be shown. Moreover, representative IHC images of staining intensity (from 0 to 3) must be shown  

"Analysis of the TROP2 expres- 228 sion in these cells showed that Eso26 homogenously express high levels of TROP2 229 mRNA and protein, while Flo1 lacks TROP2 expression": the statistically significant differences must be shown in the figure 1A with asterisks.

Author Response

Point-by-point answers to Reviewer-Comments: Revised Manuscript: “Trophoblast cell surface antigen 2 (TROP2) as a predictive biomarker for the therapeutic efficacy of sacituzumab govitecan in adenocarcinoma of the esophagus”

We would like to thank our reviewers for their helpful comments and suggestions for improvements. We have made every effort to answer your questions point by point and have repeatedly included aspects in the manuscript. Our answers and additions to the manuscript are highlighted in yellow. We have clarified the primary figures and added various other figures (supplementary figures 1-5). We are convinced that our manuscript could be significantly improved.

Reviewer 2:

  • Although the manuscript is clear and generally well written, it presents some important points that must be improved before publication. In particular:

Introduction: The introduction on TROP2 must be implemented. In particular, authors must introduce the function of this protein and specify that it is not only expressed in esophagus adenocarcinoma but also in other types of cancers. Moreover, TROP2 is also expressed in non-cancerous tissues.

Thank you very much for your comment. We have now improved the introduction by these points and considered further literature references. 

3.2. TROP2 is expressed by the majority of EAC tumors: representative images of TROP2 expression in EAC must be shown. Moreover, representative IHC images of staining intensity (from 0 to 3) must be shown  

We have now considered additional photographs documenting TROP2 expression at varying levels of expression (Supplementary Figure 5).

"Analysis of the TROP2 expression in these cells showed that Eso26 homogenously express high levels of TROP2 229 mRNA and protein, while Flo1 lacks TROP2 expression": the statistically significant differences must be shown in the figure 1A with asterisks.

Significance levels are now indicated in Fig. 1A.

Reviewer 3 Report

This manuscript stated the ADC compound Sacituzumab govitecan therapy in the esophageal adenocarcinoma. They collected 598 human EACs and found most of the patients with a high expression of Trop2. Then they employed 3 cell lines with different expression levels of Trop2 which showed the different sensitivity of SG in these cell lines.They concluded that SG could be used as a potential drug for the therapy of EAC, and Trop2 could be a biomarker of EAC.

Comments on this manuscript,

1.Some small errors should be carefully checked, like line 101 2.500 cells or 2500.

2.The curve looks weird in figure 1c, in the low concentration of SG, eg, 1 nM, the OACM5 cell seems like more resistance compared with the FLO-1 cell, and in the high concentration, more sensitive than the  eso26 cells, why?

3.Label the scale bar of the images in figure 1B.

4. We don’t see any figures of the TROP2 expression in patients by immunohistochemistry. Some representative pictures should be provided,for example the low expression middle expression and high expression of trop2.

5.In vitro study, only cell viability data showed the effect of SG in EAC. Some other assays may be provided to provide more convincing evidence, e.g. colony formation assay cell growth assay etc.

6.In vivo study, is there any toxicity of the SG? The mice body weight could be provided to show there is no toxicity.

7. The FLO-1 tumor grows very slow in the early 60 days but very fast in the later 10 days? It’s weird.

8.The SG could degrade the Trop2 protein expression or only bind it and mediate the killing effects?

9. You mentioned the total treatment time was 10 days and twice a week for the treatment, but it looks like in the figure2, after 10 days the tumors were collected?

10. Rescue study could be added for example in the low expression FLO-1 overexpress Trop2 and in the ESO26 knocking down the Trop2 to see the sensitivity of SG.

Author Response

Point-by-point answers to Reviewer-Comments: Revised Manuscript: “Trophoblast cell surface antigen 2 (TROP2) as a predictive biomarker for the therapeutic efficacy of sacituzumab govitecan in adenocarcinoma of the esophagus”

We would like to thank our reviewers for their helpful comments and suggestions for improvements. We have made every effort to answer your questions point by point and have repeatedly included aspects in the manuscript. Our answers and additions to the manuscript are highlighted in yellow. We have clarified the primary figures and added various other figures (supplementary figures 1-5). We are convinced that our manuscript could be significantly improved.

Reviewer 3:

This manuscript stated the ADC compound Sacituzumab govitecan therapy in the esophageal adenocarcinoma. They collected 598 human EACs and found most of the patients with a high expression of Trop2. Then they employed 3 cell lines with different expression levels of Trop2 which showed the different sensitivity of SG in these cell lines. They concluded that SG could be used as a potential drug for the therapy of EAC, and Trop2 could be a biomarker of EAC.

Comments on this manuscript,

1.Some small errors should be carefully checked, like line 101 2.500 cells or 2500.

2500

2.The curve looks weird in figure 1c, in the low concentration of SG, eg, 1 nM, the OACM5 cell seems like more resistance compared with the FLO-1 cell, and in the high concentration, more sensitive than the eso26 cells, why?

The curves of Eso26 and Flo-1 indeed does not show the classical sigmoid relationship between drug concentration and viability. This might suggest that other factors in these cell lines contribute to the effect of SG. This might be factors that modulate the conformation or accessibility so that we see the sum of several drug response curves. At this point, we do not know the exact mechanism that explains the shapes of the response curves.

3.Label the scale bar of the images in figure 1B.

Images are now assigned to the corresponding magnification

  1. We don’t see any figures of the TROP2 expression in patients by immunohistochemistry. Some representative pictures should be provided,for example the low expression middle expression and high expression of trop2.

We have now considered additional photos documenting TROP2 expression at different levels of expression (Supplement Figure 5).

5.In vitro study, only cell viability data showed the effect of SG in EAC. Some other assays may be provided to provide more convincing evidence, e.g. colony formation assay cell growth assay etc.

Thank you for this important point. We focused on cell culture analysis and mouse experiments in this case for the following reasons: Adenocarcinomas of the esophagus are sensitive in principle to the active ingredient of irinotecan SN-38, which is also used as the active ingredient of sacituzumab govitecan (SG). Thus, we focused on demonstrating that the specific composition of sacituzumab govitecan as an antibody-drug conjugate that binds to TROP2 on the tumor cell can also be effective in esophageal adenocarcinoma. That the prerequisite, the presence of TROP2 on the tumor cells in EAC is given and that in a mammalian model after drug application a therapeutic response can indeed be shown and that the response correlates with the presence of the target antigen. Since SG has already been successfully used in several studies (e.g. ASCENT Study in breast carcinoma), we immediately focused on a mammalian model.

  1. In vivo study, is there any toxicity of the SG? The mice body weight could be provided to show there is no toxicity.

Thank you very much for rising this issue. In the frame of health monitoring, we routinely determine the body weight of mice. Please find these data in Supplement Figure 2. We have not detected any toxicity.

  1. The FLO-1 tumor grows very slow in the early 60 days but very fast in the later 10 days? It’s weird.

We fully agree, the FLO-1 cell line has a special delayed growth behavior with rapid progression compared to the early onset Eso26 with a similar rapid progression. The difference in tumor onset between these two cell lines might be affected by their origin. FLO-1 was generated from a poorly differentiated T2 EAC and Eso26 from a poorly differentiated T4. Nevertheless, the delayed tumor onset followed by rapid tumor growth of FLO-1 is in concordance to the literature. In the study of Lui DS and colleagues (“Novel metastatic models of esophageal adenocarcinoma derived from FLO-1 cells highlight the importance of E-cadherin in cancer metastasis”; Oncotarget. 2016; 7:83342-83358) tumor growth of the parental FLO-1 starts lately, around day 80 followed by a rapid increase in tumor mass. We have taken this important point into account in the results section.

8.The SG could degrade the Trop2 protein expression or only bind it and mediate the killing effects?

Indeed, according to current knowledge, tumor cell toxicity occurs via two pathways: a) via a TROP2 binding of SG followed by an internalization of SG and b) due to a weak linkage of SN38 with the antibody to a release of SN38 into the pericellular space with toxic local effect. Whether there is actually a significant downregulation of TROP2 on the carcinoma cell in this context is not reliably described and we do not see it to a significant extent.

  1. You mentioned the total treatment time was 10 days and twice a week for the treatment, but it looks like in the figure2, after 10 days the tumors were collected?

Thank you for mentioning this unclear description. We included the individual time points of therapy application in the methods and materials sections 2.5. In vivo xenograft treatment, as the following: Mice were treated on day 0 (therapy initiation), day 3, day 7 and day 10 with 25 mg/kg/body weight 129 (BW) Trodelvy (Sacituzumab govitecan-hziy, Immunomedics Inc., USA) or with solvent- 130 control by i.p. injection in concordance with the established protocol for breast cancer xenograft treatment.

  1. Rescue study could be added for example in the low expression FLO-1 overexpress Trop2 and in the ESO26 knocking down the Trop2 to see the sensitivity of SG.

We agree that rescue experiments would provide additional confidence in our interpretation. Unfortunately, addressing this issue is out of the approval of our animal experiment licence. We feel that our findings in the context of prior data on the mechanism of action of SG through TROP2 and clinical data is supporting our findings. We hope that reviewer can agree.

Round 2

Reviewer 1 Report

I appreciate the author's efforts in their responses and agree with their changes.  

Reviewer 2 Report

the manuscript has been significantly improved and can be accepted in the present form.

Reviewer 3 Report

Thanks you for providing the supplementary figures for the review.

From my perspective, I have no other questions about this manuscript.